# Experimental Study of Geysering in an Upstream Vertical Shaft

**Daniel G. Allasia** [1], **Liriane Élen Böck** [1], **Jose G. Vasconcelos** [2,*], **Leandro C. Pinto** [1], **Rutineia Tassi** [1], **Bruna Minetto** [1], **Cristiano G. Persch** [1] **and Robson L. Pachaly** [2]

[1] Department of Sanitary and Environmental Engineering, Universidade Federal de Santa Maria, Santa Maria 97105-900, RS, Brazil; dallasia@gmail.com (D.G.A.); leandro.pinto@ufsm.br (L.C.P.); ruti@ufsm.br (R.T.); bruna.minetto@gmail.com (B.M.)

[2] Department of Civil and Environmental Engineering, Auburn University, Auburn, AL 36849-5337, USA; rlp0046@auburn.edu

\* Correspondence: jgv@auburn.edu; Tel.: +1-334-844-6280

**Abstract:** Transient flows in stormwater systems can lead to damaging and dangerous operational conditions, as exemplified by geysering events created by the uncontrolled release of entrapped air pockets. Extreme rain and associated rapid inflows may result in air pocket entrapment, which causes significant changes in flow conditions and potentially geysering. Stormwater geysers have been studied in different experimental and numerical modeling studies, as well as from limited data gathered within real systems. However, there is still no complete understanding of geysering events, as stormwater system geometries vary considerably. Most past studies involved releasing air from an intermediate shaft, in which a significant fraction of the entrapped air may bypass the release. This work advances the understanding of geysering by considering uncontrolled air release through an upstream shaft or manhole. In such cases, the entire air pocket is released upon reaching the shaft, worsening the occurrence of geysering. Pressure and water level measurements were performed for various combinations of initial water pressure, trapped air pocket volume, and vertical shaft geometries, indicating the higher severity of these geysering events. The results obtained also corroborate previous studies in that the measured pressure heads were lower than the grade elevation. Future studies should include larger-scale testing and the representation of this geometry using CFD.

**Keywords:** stormwater systems; air pockets; sewer pressurization; two-phase flows

## 1. Introduction and Objectives

Intense rain events leading to rapid filling conditions of stormwater collection systems may create operational problems such as pressurization of conduits, entrapment of air pockets within the pipes, pressure surges, and even water hammer [1–4]. Large entrapped air pockets spread and move within stormwater conduits as gravity currents [5–7], and as they reach the bottom of water-filled ventilation, uncontrolled air release ensues. Entrapped air will move upwards due to buoyancy, and as it moves, it displaces the water initially present in the structure. When the displaced water reaches grade elevation, an explosive release of an air–water mix is observed, referred to as stormwater geysers. Stormwater geysering events have been investigated as a single-phase flow phenomenon since the 1990s [8]. Still, two-phase flow geysers have been investigated more recently [4,9] and, since then, have been studied by many authors. In some episodes, stormwater geysers were reported to reach more than 30 m [10]. These events can negatively impact the environment and public health and create additional costs related to the damage of the drainage infrastructure [1,4,8,11,12]. The severity of the observed geysering events is more consistent with two-phase flow interactions than with single-phase mass oscillation phenomena [13]. As such, research continues to be devoted to air entrapment and release processes in stormwater systems, as exemplified in [10,14,15].

Past contributions included experimental investigations [9,16–19], numerical investigations [14,15,19–22], and, more rarely, field observations [3]. The severity of geysering is

linked to some extent to the volume of the released air pocket, as smaller air pockets do not appear to create issues. The air pocket displacement appears to be impacted by the initial pressure once the air volume expands as it is rapidly depressurized during its release [11]. Experimental observations of the vertical air–water flow during the rise of the entrapped air pocket indicate the existence of an upper water interface in contact with atmospheric air. Underneath is a rising air–water interface shaped as a Taylor bubble advance, often more rapidly than the upper interface at the shaft's top [4]. The upper interface may reach grade elevation, and the upward-moving water impact can damage shafts or manhole structures [23]. The geysering intensity is comparatively smaller up to the moment when the rising air pocket reaches grade [19], at which point the large pressure gradient creates very large air–water velocities exceeding 24 m/s [23]. Observed mist in geysers can reach very high velocities and can be attributed to air–water shear in the vertical riser, similar to flood instability phenomena [24]. While other geysering mechanisms can exist, the rise and severity of some observed geysering events can be attributed to the uncontrolled release of entrapped air pockets.

Various past investigations assessed the effects of system geometry on the intensity of geysering. In particular, some studies considered the effects of the diameter of the water-filled vertical structures and the height of the standing water, as exemplified in [4,18]. Mechanisms for geyser mitigation that involved shaft geometry changes were presented in [9,25], including increasing the vertical shaft diameter or adding a chamber near grade, with some success in decreasing geysering strength. In summary, it is acknowledged that system geometry is an important factor to be considered in studies that involve stormwater geysering.

However, an important knowledge gap still exists regarding the effects of the system geometry concerning geysering events created by the uncontrolled release of entrapped air pockets. To the authors' knowledge, no previous studies considered a geometry in which the air pocket is eliminated through an upstream shaft instead of an intermediary release point. As pointed out in the numerical results by [11], a vertical shaft is captured between 11% and 70% of the initial air pocket mass before geysering initiates. Thus, a significant fraction of the air pocket mass might move downstream and not contribute to geysering in an intermediate vertical structure. By contrast, it is hypothesized that an upstream release point would experience more severe geysering since there would be no other location for air to be vented. In addition, such geometry was never considered in past experimental studies concerning geysering.

A second knowledge gap identified in our literature review is linked with the method used for air admission in these studies. Most geysering experimental studies (e.g., [3,4,9,18]) considered a sudden opening of a valve or had rapid releasing air into the pressurized water flow. As is well-known, such sudden release creates air–water surging characterized by pressure oscillations in the system before geysering. The effects of gradual air pressurization and subsequent release have not yet been investigated experimentally.

This work addresses these knowledge gaps by performing systematic studies and presenting results obtained in a large-scale experimental apparatus. This work initially presents the methodology (Section 2) used in the experimental tests, including the experimental procedures, measured variables, and instrumentation. The results are subsequently presented and discussed in Section 3, including comparing new measurements and those from previous investigations. Finally, conclusions and recommendations for future investigations are presented.

## 2. Materials and Methods

### 2.1. Experimental Apparatus

The experimental apparatus has features that are similar to the upstream portion of a stormwater system undergoing pressurization, characterized by a water-filled upstream dropshaft. The apparatus, which represents a simplification of an actual stormwater system, also had a lower connection linked to a reservoir to sustain the system pressure during the

air pocket release (following [9]), and an air chamber in which air was initially stored, as shown in Figure 1.

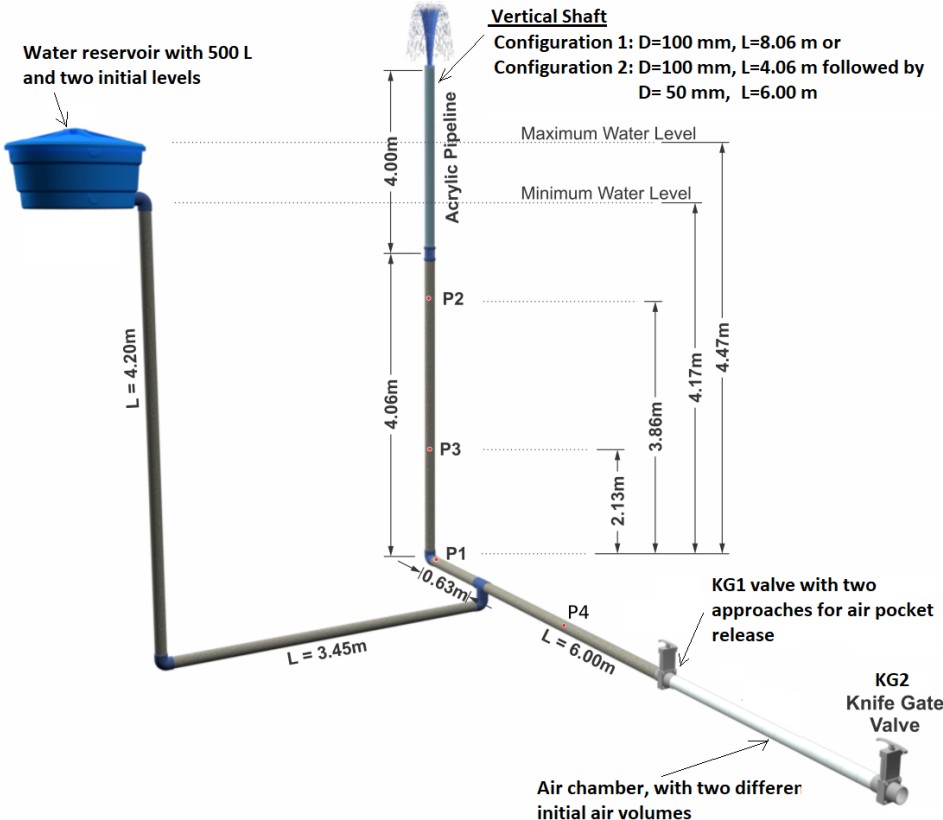

**Figure 1.** Sketch of experimental apparatus used for this research. The 100 mm vertical shaft configuration is shown.

The apparatus included a ∼7 m long horizontal PVC pipeline, with internal diameter $D_t$ equal to 100 mm. The apparatus was fitted with an air chamber that was separated from the remainder of the apparatus by a knife gate valve (KG1) with the same diameter $D_t$. The maneuver, rapid or gradual, of valve KG1 released the air pocket into the water-filled apparatus. On the other extreme of the air chamber, another knife gate valve (KG2) was used to drain the air chamber and admit atmospheric air before each experimental run. Six meters from the air chamber, there was a T-junction with a downward derivation that was connected through a 3.45 m horizontal pipe to a 500 L reservoir filled with water, which was placed at a much higher elevation. This reservoir was present to sustain pressure during the air release [9], and the downward connection prevented the released air pocket from escaping through the T-junction. Two elevations were considered in this reservoir: Higher-level (referred to as HL) had an initial piezometric head elevation of 4.47 m above the KG1 valve, and Low-level (LL) had a piezometric head elevation 4.17 m above the KG1 valve.

A 90-degree elbow was placed 0.63 m from the T-junction, allowing the connection of the horizontal piping system to the water tower, which represents a terminal dropshaft, this being the only location in the system where the entrapped air pockets were released. Two different geometries for this dropshaft were used. The first geometry was an 8.06 m long, fixed-diameter tower with diameter $D = D_t = 100$ mm, as shown in Figure 1. This tower's top 4.0 m segment was fit with acrylic pipe walls, which enabled visual tracking of the rising free surface and air pocket interface during air pocket releases. The second geometry had the same initial segment with 4.06 m long and $D = 100$ mm, but had a second segment with a length of 6.0 m and diameter of 50 mm, for a total length of 10.06 m. This second tower was entirely constructed in PVC. Thus, the tracking of the initial advance

of the free surface was attained by drilling small lateral holes in the tower (placed every 0.50 m), which discharged water as the internal water level rose. Before the air pocket release, the water level in the towers matched the water level of the 500 L reservoir, given that the conditions were quiescent before the air pocket release.

### 2.2. Data Acquisition

A set of four pressure transducers was used to assess the pressure head in the experimental apparatus, and Figure 1 presents the locations where pressure measurements were taken. Pressure transducer P1 was placed at the base of the vertical tower, and P2 was placed 3.86 m above. These two were Druck UNIK 5000 transducers with a pressure range from $-34.47$ to 68.95 kPa ($-5$ to 10 PSI) with a precision of $\pm0.04\%$ of the full scale ($+/-41$ Pa or 4 mm of pressure head). These two transducers were connected to a National Instruments NI-USB 6210 data acquisition board, and samples were taken at a frequency of 300 Hz. Two other transducers (P3 and P4) were used in the apparatus. Transducer P3, model MPX 5050 DP, by NXP, was placed 2.13 m into the vertical tower, and transducer P4, model MPX 5100 DP, by NXP, was placed approximately 3 m away from valve KG1 in the horizontal pipe. Transducers P3 and P4 have pressure ranges of 50 kPa and 100 kPa, respectively, both with an accuracy of 300 Pa ($\sim0.03$ m), sampling at a smaller frequency of 10 Hz, and connected to an Arduino Uno datalogger. To monitor the water level at the top reservoir, an ultrasonic ranging sensor US (HC–SR04) with an accuracy of 0.03 m was installed. A Samsung Note 20 and S9+ video-recording cameras were used to track the advance of the free surface and air–water interfaces at 30 frames per second.

### 2.3. Experimental Procedure

The experimental procedure involved the following steps:

1. KG1 was closed and the apparatus was filled up to a pre-determined piezometric head in the reservoir, either the High-level (HL) or the Low-level (LL) elevations in the reservoir.
2. Water inflow was stopped to enable the system to reach quiescent conditions.
3. KG2 was opened and later closed e to admit the atmospheric air into the air chamber.
4. Data acquisition was initiated from the pressure transducers and video cameras.
5. KG1 was quickly or gradually opened, as described below.
6. The release of the air pocket through the vertical tower was monitored, which typically started 15 s after the total opening of KG1.
7. Data acquisition was stopped and the tested condition was repeated (at least three times) to ensure consistency of the collected data. These repetitions are referred to as R1, R2, and R3.

Two opening conditions for the KG1 valve were considered. The first condition is referred to as Quick Valve Opening (Q), where KG1 was manually opened in less than 0.5 s. This opening immediately released the apparatus's air pocket and created air–water surging, similar to the conditions described by [11]. This surging was reflected in significant pressure oscillations, as will be shown in the following results, which occurred as the air pocket was released. The second type of opening, referred to as Gradual Valve Opening (G), involved the air pocket release in two stages. In the first stage, KG1 was opened partially (about 10% of the diameter) to enable the pressurization of the air phase prior to the release of the air into the apparatus while preventing air release into the apparatus. About 20 s after the partial opening and after mitigating air–water surging, KG1 was fully opened and the pressurized air pocket was released into the apparatus. This was an important change from previous experimental research since the initial air phase pressure was atmospheric, which was significantly smaller than the water pressure. Such an approach enabled a better visualization of the effects of the air phase release since the effects of the initial air–water surging were mostly absent. Following the entrapment of air, the process of air pressurization should be gradual following the pressurization of water flows. Thus, the absence of surging prior to the uncontrolled release of air pockets is more representative.

*2.4. Experimental Variables*

Referring to Figure 1, this experimental apparatus considered four variables yielding 16 unique experimental conditions:

a    Two vertical shaft configurations: Configuration 1 used an 8.06 m long shaft with diameter $D$ = 100 mm. Configuration 2 used a shaft with an initial diameter of $D$ = 100 mm and length of 4.06 m followed by a segment with $D$ = 50 mm with a length of 6.0 m.

b    Two initial piezometric heads were imposed in the 500 L water reservoir: either 4.47 m or 4.17 m, measured from the invert of the horizontal pipe. These conditions are also referred to as HL and LL, respectively.

c    Two air pocket volumes of $\forall$ = 25 L and 45 L, in both cases in atmospheric pressure.

d    Two approaches for opening the KG1 valve and releasing the air phase: either a Quick (Q, within 0.5 s) or gradual (G).

Normalized variables were used to compare the present work's results with results from different studies. The dropshaft/vertical tower diameter, referred to as $D$, was normalized by the horizontal tunnel diameter $D^* = D/D_t$. The initial water free surface coordinate $Y_{FS,0}$ was normalized by the length of the vertical tower $Y_{FS,0}^* = Y_{FS,0}/L_s$, and the corresponding vertical displacements $\Delta Y_{FS}$ were normalized by the initial depth of the water in the tower $\Delta Y_{FS,0}^* = \Delta Y_{FS}/Y_{FS,0}$. Velocities of the free surface $V_{FS}$ were normalized as $V^* = V_{FS}/\sqrt{gD}$, where $g$ is the gravitational acceleration.

Two different approaches were used to normalize the released air pocket. The first type is a normalization based on the $D_t$ ($\forall^* = \forall/D_t^3$) used in most past experimental contributions involving the release of an entrapped air pocket. The second normalization is a newly proposed normalization used in this research. The air pocket volume is normalized by the initial volume of the water present in the dropshaft, i.e., $\forall^{**} = 4\forall/(Y_{FS,0}\pi D^2)$.

A list of the tested experimental conditions is presented in Table 1. Experimental conditions were repeated between 3 and 6 times to ensure consistency of the experimental results. Overall, minor differences in the results were verified for the same experimental conditions during the runs performed. Normally, the experimental repetitions were stopped in the third run (R1, R2, and R3); additional runs were performed either when inconsistent results were observed or due to operational problems during a given run.

**Table 1.** Normalized experimental conditions used in the present research.

| No. Tested Condition | Valve Opening | $D^*$ | $Y_{FS,0}^*$ | $\forall^*$ | $\forall^{**}$ |
|---|---|---|---|---|---|
| C1 | G | 0.50 | 0.44 | 25.0 | 2.85 |
| C2 | Q | 0.50 | 0.44 | 25.0 | 2.85 |
| C3 | G | 0.50 | 0.44 | 45.0 | 5.13 |
| C4 | Q | 0.50 | 0.44 | 45.0 | 5.13 |
| C5 | G | 1.00 | 0.56 | 25.0 | 0.71 |
| C6 | Q | 1.00 | 0.56 | 25.0 | 0.71 |
| C7 | G | 1.00 | 0.56 | 45.0 | 1.28 |
| C8 | Q | 1.00 | 0.56 | 45.0 | 1.28 |
| C9 | G | 0.50 | 0.42 | 25.0 | 2.85 |
| C10 | Q | 0.50 | 0.42 | 25.0 | 2.85 |
| C11 | G | 0.50 | 0.42 | 45.0 | 5.13 |
| C12 | Q | 0.50 | 0.42 | 45.0 | 5.13 |
| C13 | G | 1.00 | 0.52 | 25.0 | 0.71 |
| C14 | Q | 1.00 | 0.52 | 25.0 | 0.71 |
| C15 | G | 1.00 | 0.52 | 45.0 | 1.28 |
| C16 | Q | 1.00 | 0.52 | 45.0 | 1.28 |

Table 2 presents selected experimental and numerical conditions from [11] that are used to compare the results.

**Table 2.** Selected experimental conditions presented in the research by [11] for comparison with the present research.

| No. Tested Condition | Valve Opening | $D^*$ | $Y^*_{FS,0}$ | $\forall^*$ | $\forall^{**}$ |
|---|---|---|---|---|---|
| M1 | Q | 0.34 | 0.55 | 3.63 | 4.53 |
| M2 | Q | 0.51 | 0.55 | 3.63 | 2.01 |
| M3 | Q | 0.67 | 0.55 | 3.63 | 1.16 |
| M4 | Q | 0.34 | 0.55 | 7.26 | 9.07 |
| M5 | Q | 0.51 | 0.55 | 7.26 | 4.03 |
| M6 | Q | 0.67 | 0.55 | 7.26 | 2.31 |
| M7 | Q | 0.50 | 0.33 | 5.00 | 0.51 |
| M8 | Q | 1.00 | 0.33 | 5.00 | 0.13 |
| M9 | Q | 0.50 | 0.33 | 10.0 | 1.02 |
| M10 | Q | 1.00 | 0.33 | 10.0 | 0.26 |
| M11 | Q | 1.00 | 0.33 | 50.0 | 1.27 |

*2.5. Data Analysis*

Once the experimental runs were complete, the data analysis process was initiated. Videos were downloaded and visually analyzed to determine the vertical displacement of the water's free surface within the vertical shaft ($\Delta Y_{FS}$). This was performed by analyzing the frame-by-frame progression, identifying the time in which each surface reached the marks on the vertical axis. These data were then used to calculate the velocity associated with $\Delta Y_{FS}$ displacements by a second-order finite difference scheme.

## 3. Results and Discussion

*3.1. Free Surface Water Results*

The experimental results were consistent with the existing literature in that releasing the entrapped air pocket through the water-filled vertical shaft created a displacement of the free surface. Results for the free surface displacement and free surface velocity for $D^* = 1.0$ and all combinations of pocket volume, air valve opening strategy, and initial water level are presented in Figure 2. The same type of results, albeit for the cases when $D^* = 0.5$, are presented in Figure 3, and all these results are summarized in Table 3. The following observations can be drawn from these measurements:

- The maximum normalized free surface displacements $\Delta Y^*_{FS,max}$ for the cases with $D^* = 1.0$ were in the range from 0.42 to 0.67 (Figure 2 and Table 3). There were some cases with oscillations during the free surface rising, and, in other cases, a quasi-monotonic rising was observed.
- The results with $D^* = 0.5$ and shown in Figure 3 consistently exceeded the length of the vertical shaft (i.e., $\Delta Y^*_{FS} > 1.4$), creating a geysering. The free surface rising had an early stage of gradual change, followed by a very steep rise in the last two seconds of the air pocket release. The results agree with past observations that reported that smaller-diameter shafts are more prone to geysering.
- The normalized free surface velocity values fluctuated significantly for the cases with $D^* = 1.0$, as shown in Figure 2. The oscillation range was observed for the case with the largest air pocket volume and quick opening of the KG1 valve. In a few cases, the free surface velocities even briefly became negative. Over the air pocket release duration, the average normalized velocity was 0.2 to 0.3, with a maximum of 0.8.
- By contrast, for $D^* = 0.5$, the normalized free surface velocity remained positive, as shown in Figure 3. A first, longer phase of smaller free surface velocity is succeeded by a very fast air pocket release, with a normalized velocity that often exceeded 10, thus much faster than the conditions with larger shaft diameter.

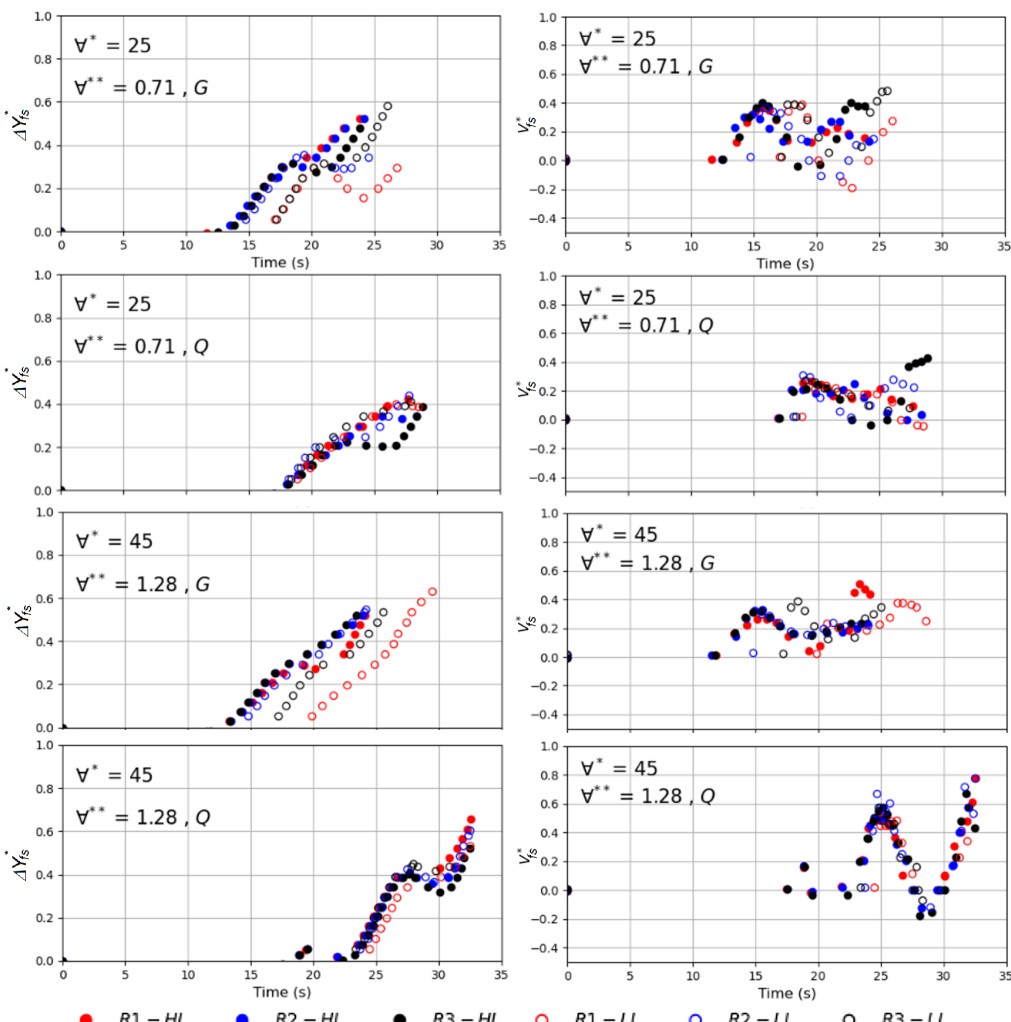

**Figure 2.** Time evolution of normalized water free surface displacement (**left**) and velocity (**right**) for all tests with $D^* = 1.0$. R1, R2, and R3 represent the repetitions, while HL and LL represent high and low initial piezometric heads in the reservoir.

Furthermore, as hypothesized, the air pocket released through a terminal dropshaft created more severe geysering conditions. Taking the experimental and numerical conditions presented in [11] and summarized in Table 4, the following observations are made:

- The measured $V^*_{FS,max}$ presented in [11] for $D^* = 0.34$ and 0.51 (conditions M1, M2, M4, and M5 in Table 4) are compared with the ones obtained in the present work for $D^* = 0.50$. It can be noticed that the velocity values are systematically two to three times larger in the present experiment. While we attribute this to the geometry of the single dropshaft, another factor that could contribute to this difference is the vertical length of the tower in the present experiment. The vertical shaft presented in [11] had a length under 5 m, whereas it was over 10 m long in the present work.

- The measured $\Delta Y^*_{FS,max}$ presented in [11] for $D^* = 0.67$ (conditions M3 and M6 in Table 4) is compared with the ones obtained in the present work for $D^* = 1.0$. Even though the value used in the present experiment for $D^*$ is much larger than the conditions in [11], the reported free surface displacement varied from 0.56 to 0.81, which is in the range of the variation in the present study, from 0.42 to 0.67.

- The $V^*_{FS,max}$ from CFD modeling results shown in [11] for $D^* = 0.5$ and $Y^*_{FS,0} = 0.33$ (conditions M7 and M9) are compared with the cases in the present study for the same $D^*$. Again, the normalized maximum velocity values in the present work are much larger than the modeling results in [11].

- The $\Delta Y^*_{FS,max}$ from CFD modeling results shown in [11] for $D^* = 1.0$ and $Y_{FS,0^*} = 0.33$ (conditions M8, M10, and M11) are compared with the cases in the present study for the same $D^*$. The normalized displacements measured in the CFD analysis varied from 0.20 to 0.45 depending on the air pocket volume. In the present work, for smaller air pocket volumes, $\Delta Y^*_{FS,max}$ varied from 0.42 to 0.60, and from 0.57 to 0.67 for larger pockets. It is important to reiterate that, as presented in [11], the vertical shaft did not release the entire air pocket, unlike the conditions in the present study.

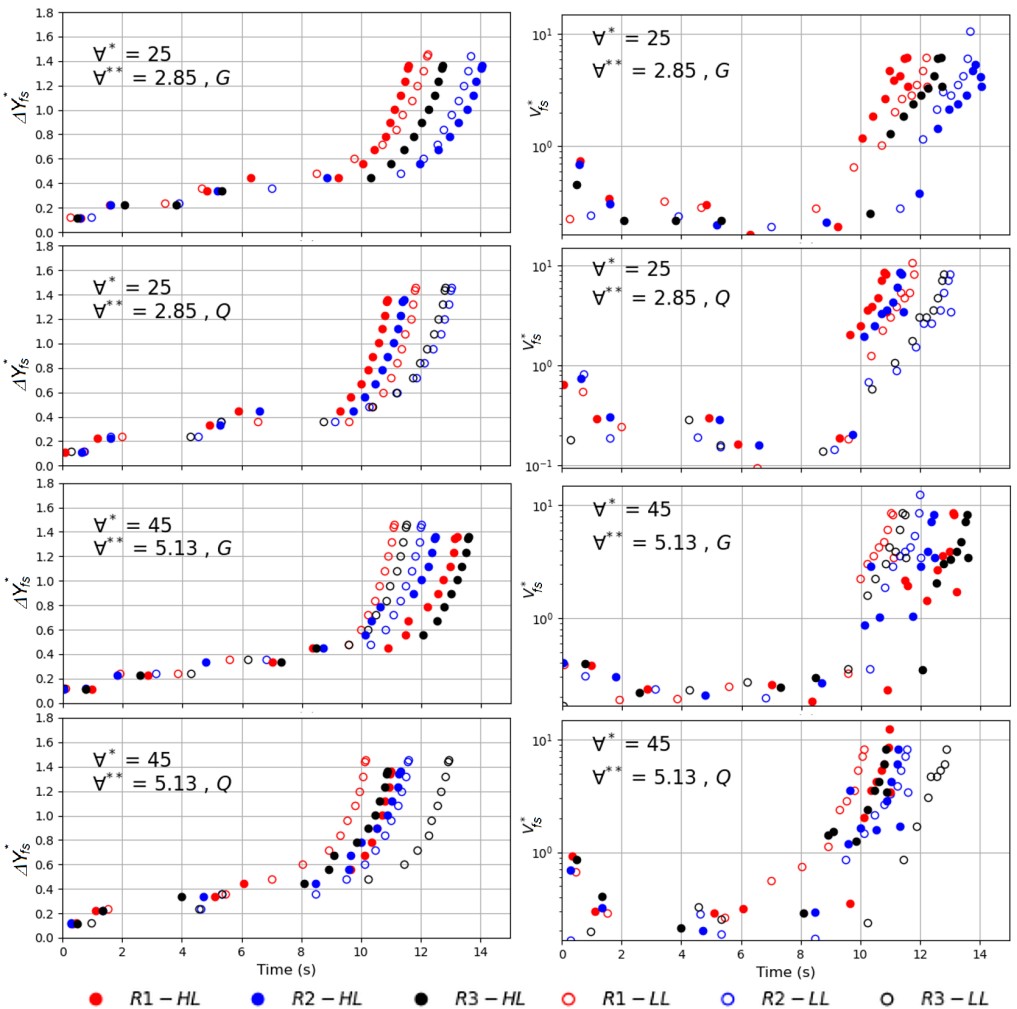

**Figure 3.** Time evolution of normalized water free surface displacement (**left**) and velocity (**right**) for all tests with $D^* = 0.50$.

Figure 4 presents results for $\Delta Y^*_{FS,max}$ versus the two approaches of normalized volume. In the left graph of Figure 4, it is possible to clearly observe the shaft diameter effect in the free surface displacement when plotted against $\forall^*$. Such an effect is not observed for the lower air pocket volume conditions ($\forall^* < 10$). When results are plotted against $\forall^{**}$, a collapse could be observed, grouping the $Y^{**}_{FS,0} > 80$ results for higher values of $\Delta Y_{FS,0}$, while lower values are grouped for $Y^{**}_{FS,0} < 50$. Here $Y^{**}_{FS,0}$ corresponds to the initial free surface coordinate $Y_{FS,0}$ normalized by the dropshaft diameter $D$. These results show that there is an important dependence of the free surface elevation in the dropshaft to the initial piezometric head. The grey shadow mark in Figure 4 is a prospection for a map separating zones of curves for free surface displacement as a function of $D^*$ and $Y^{**}_{FS,0}$. More tests must be carried out to better delimit such regions in the map.

**Table 3.** Selected experimental results for $\Delta Y^*_{FS,max}$ and $V^*_{FS,max}$ from present research.

| Tested Condition | Valve Opening | $D^*$ | $Y^*_{FS,0}$ | $\forall^*$ | $\forall^{**}$ | $\Delta Y^*_{FS,max}$ | $V^*_{FS,max}$ |
|---|---|---|---|---|---|---|---|
| C1 | G | 0.50 | 0.44 | 25.0 | 2.85 | >1.36 | 6.2 |
| C2 | Q | 0.50 | 0.44 | 25.0 | 2.85 | >1.36 | 8.6 |
| C3 | G | 0.50 | 0.44 | 45.0 | 5.13 | >1.36 | 8.6 |
| C4 | Q | 0.50 | 0.44 | 45.0 | 5.13 | >1.36 | 8.3 |
| C5 | G | 1.00 | 0.56 | 25.0 | 0.712 | 0.54 | 0.45 |
| C6 | Q | 1.00 | 0.56 | 25.0 | 0.712 | 0.42 | 0.43 |
| C7 | G | 1.00 | 0.56 | 45.0 | 1.28 | 0.57 | 0.51 |
| C8 | Q | 1.00 | 0.56 | 45.0 | 1.28 | 0.67 | 0.78 |
| C9 | G | 0.50 | 0.42 | 25.0 | 2.85 | >1.45 | 11 |
| C10 | Q | 0.50 | 0.42 | 25.0 | 2.85 | >1.45 | 11 |
| C11 | G | 0.50 | 0.42 | 45.0 | 5.13 | >1.45 | 12 |
| C12 | Q | 0.50 | 0.42 | 45.0 | 5.13 | >1.45 | 8.5 |
| C13 | G | 1.00 | 0.52 | 25.0 | 0.712 | 0.60 | 0.46 |
| C14 | Q | 1.00 | 0.52 | 25.0 | 0.712 | 0.42 | 0.21 |
| C15 | G | 1.00 | 0.52 | 45.0 | 1.28 | 0.62 | 0.39 |
| C16 | Q | 1.00 | 0.52 | 45.0 | 1.28 | 0.61 | 0.78 |

**Table 4.** Selected experimental results for $\Delta Y^*_{FS,max}$ and $V^*_{FS,max}$ presented in the research by [11] for comparison with the present research.

| Tested Condition | Valve Opening | $D^*$ | $Y^*_{FS,0}$ | $\forall^*$ | $\forall^{**}$ | $\Delta Y^*_{FS,max}$ | $V^*_{FS,max}$ |
|---|---|---|---|---|---|---|---|
| M1 | Q | 0.34 | 0.55 | 3.63 | 4.53 | 0.78 | 2.7 |
| M2 | Q | 0.51 | 0.55 | 3.63 | 2.01 | 0.74 | 3 |
| M3 | Q | 0.67 | 0.55 | 3.63 | 1.16 | 0.56 | 1 |
| M4 | Q | 0.34 | 0.55 | 7.26 | 9.07 | 0.78 | 3 |
| M5 | Q | 0.51 | 0.55 | 7.26 | 4.03 | 0.74 | 3 |
| M6 | Q | 0.67 | 0.55 | 7.26 | 2.31 | 0.81 | 2.8 |
| M7 | Q | 0.50 | 0.33 | 5.00 | 0.519 | 0.45 | 2.70 |
| M8 | Q | 1.00 | 0.33 | 5.00 | 0.127 | 0.20 | 0.50 |
| M9 | Q | 0.50 | 0.33 | 10.0 | 1.02 | 1.10 | 3.20 |
| M10 | Q | 1.00 | 0.33 | 10.0 | 0.255 | 0.38 | 0.70 |
| M11 | Q | 1.00 | 0.33 | 50.0 | 1.273 | 0.45 | 1.00 |

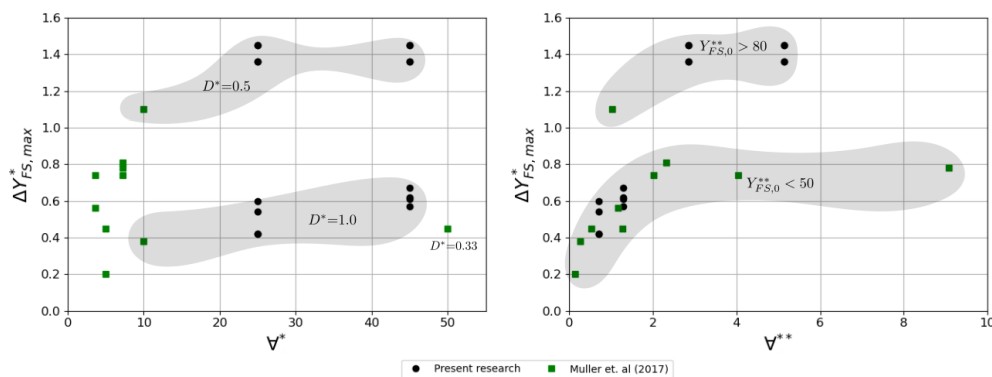

**Figure 4.** Plot of maximum normalized water free surface displacement $\Delta Y^*_{FS,max}$ versus normalized volume $\forall^*$ (**left**) and $\forall^{**}$ (**right**) obtained in this research and in [11].

### 3.2. Pressure Measurements

In addition to the results tracking the advance of the free surface flow in the vertical tower, pressure measurements were taken at various locations in the apparatus. As explained earlier, the pressure transducer nearest to the location of the air pocket release

was P4, thus experiencing the strongest air–water surging in the cases when the air pocket release through valve KG1 was quickly opened (Q cases). At various locations, the other three pressure transducers were placed near the vertical tower. As is indicated in Figure 1, transducer P1 was placed at the tower's base, and P2 was placed 3.86 m above the horizontal pipes. Finally, P3 was placed at an intermediate elevation of 2.13 m. The rationale for the transducer placement in the vertical tower was to ensure that these were underwater as the experiments would start. Likewise, in the results presented in Figures 2 and 3, time zero corresponded to the moment when the valve KG1 was completely opened, releasing the air in the apparatus.

Consistent with the findings of previous studies on geysering, the pressure head measurements were always beneath the maximum observed water elevation during the air release. Even considering the transducers experiencing the highest static pressures (i.e., P1 and P4) and the air–water surging in the cases when KG1 was quickly opened, the measured pressure never exceeded 6 m in any of the test conditions. The amplitude of the pressure fluctuation depended on the diameter of the vertical shaft and the volume of the air pocket.

As is shown in Figure 5, the pressure results with gradual opening showed much smaller oscillations than the quick opening, as would be anticipated. The gradual opening eliminated most of the effects of air–water surging. After an initial stage of minor pressure fluctuation detected in P4 and P1, there were pressure fluctuations between $T = 15$ to 30 s associated with releasing the air pocket through the vertical tower. After air release through the tower, the results from the cases with the quick and gradual opening of KG1 are quite similar.

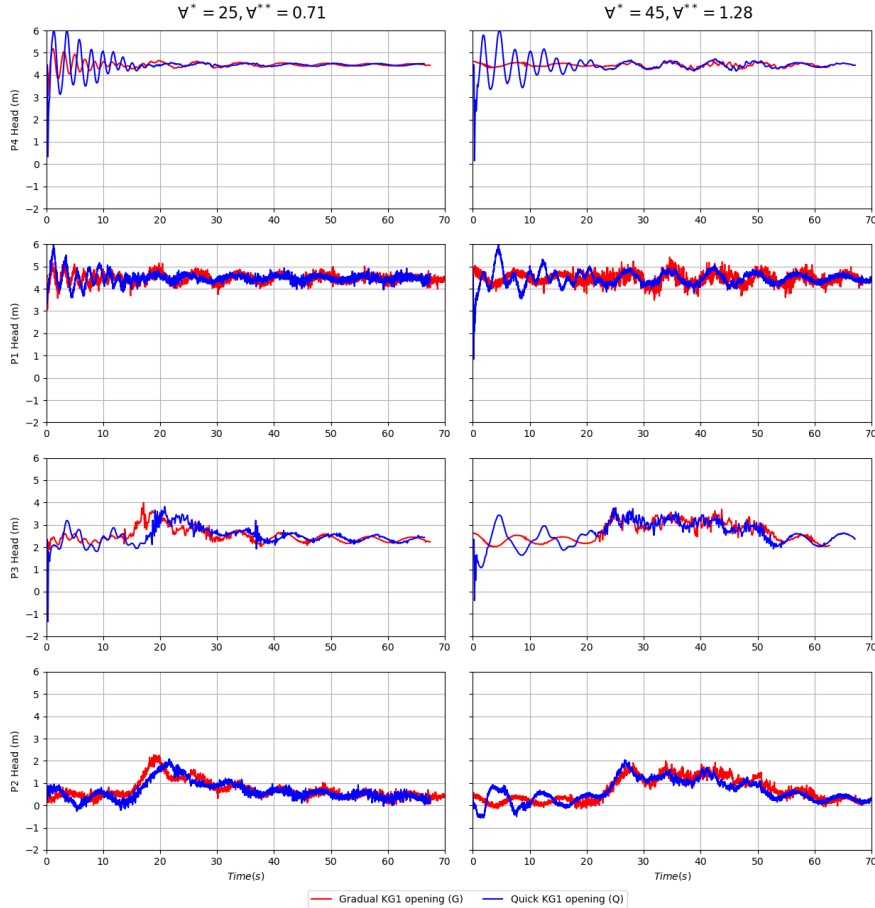

**Figure 5.** Pressure head results for each transducer for the case when $D^* = 1.0$, and higher initial piezometric level in the reservoir. Results in the left column are for the smaller air pocket, and the right column is for the larger air pocket volume.

It is also observed that for the case of $\forall^* = 45$, the pressure signals for P2 and P3 sensors are sustained at a level around 3 m and 2 m, respectively. For $\forall^* = 25$, a first pressure peak occurs followed by a gradual pressure decrease. Such results could be associated with a larger time air release in the $\forall^* = 45$ case.

The results presented in Figure 6 differ from the ones presented in Figure 5 in that the water reservoir level was smaller. The pressure oscillations for the case with the gradual opening were even smaller. The largest pressure increase for transducers P3 and P2, as shown in the case of Figure 5, occurred during the release of the air pocket through the vertical shaft. For both these cases, the pressure increase was in the same range as the rise of the water level observed in the 100 mm towers. Furthermore, it is possible to observe the sustained pressure level in transducers P3 and P2 during air release for the case with $\forall^* = 45$.

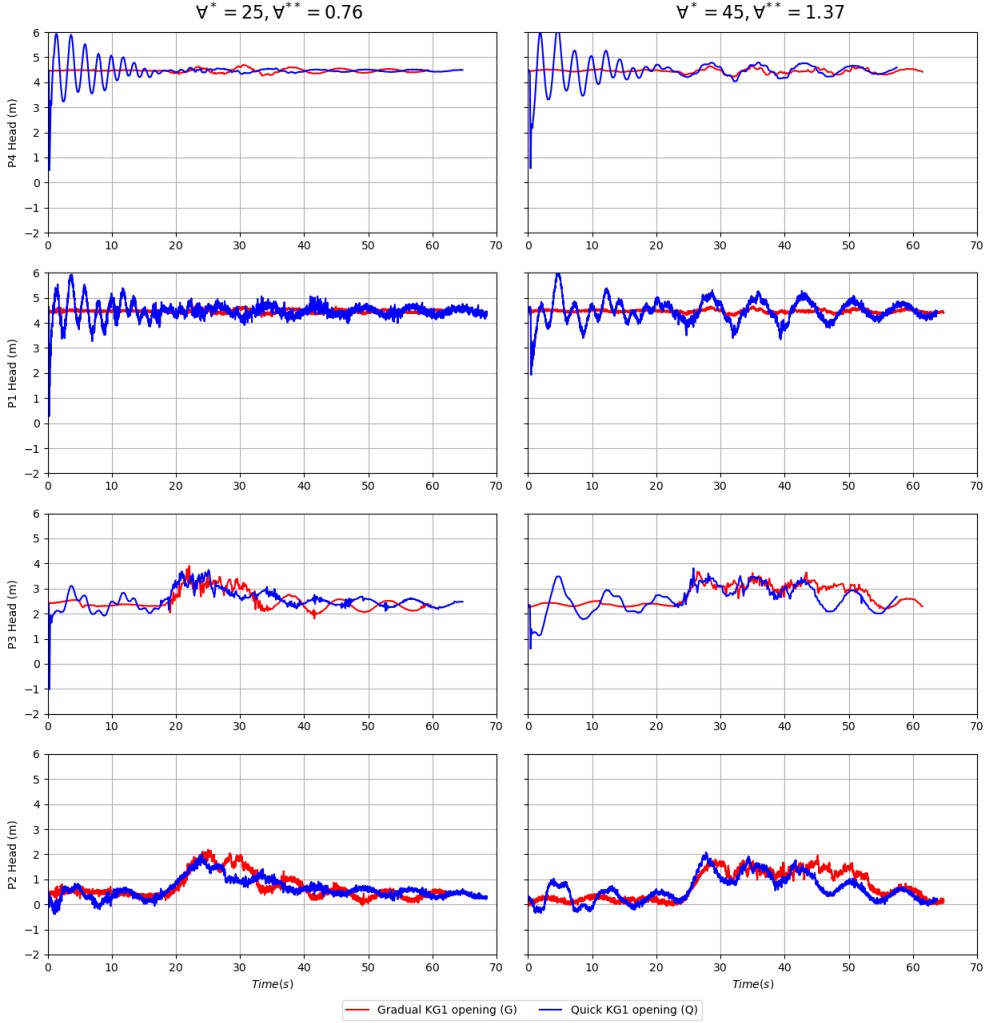

**Figure 6.** Pressure head results for each transducer for the case when $D^* = 1.0$, and lower initial water level in the water reservoir. Results in the left column are for the smaller air pocket, and the right column is for the larger air pocket volume.

Figures 7 and 8 present the pressure results for the two cases with $D^* = 0.50$ when the water reservoir level was in the high and low levels, respectively. Compared with the results obtained for $D^* = 1.0$, the amplitude associated with the air–water surging was much larger. Furthermore, the pressure fluctuations observed in transducers P2 and P3 were much larger than the ones placed at the level of the horizontal pipe. Unlike the case with the larger vertical tower diameter, the range of the pressure oscillations for the

transducer P2 exceeded the corresponding values for the transducer at a lower elevation P3. The maximum pressure in these transducers corresponded to the earlier stage of the air–water surging when KG1 was quickly opened. This demonstrates the importance of the gradual opening of the valve before releasing the air pockets, to attenuate air–water surging and thus reduce this effect in the geysering measurements. As was the case with the larger-diameter vertical tower, after the release of air in the system, the results obtained with the gradual and quick opening are similar. Finally, considering the maximum recorded pressure head, the values did not correspond to the top elevation of the vertical tower. However, the increase in the piezometric head was larger than the case with $D^* = 1.0$.

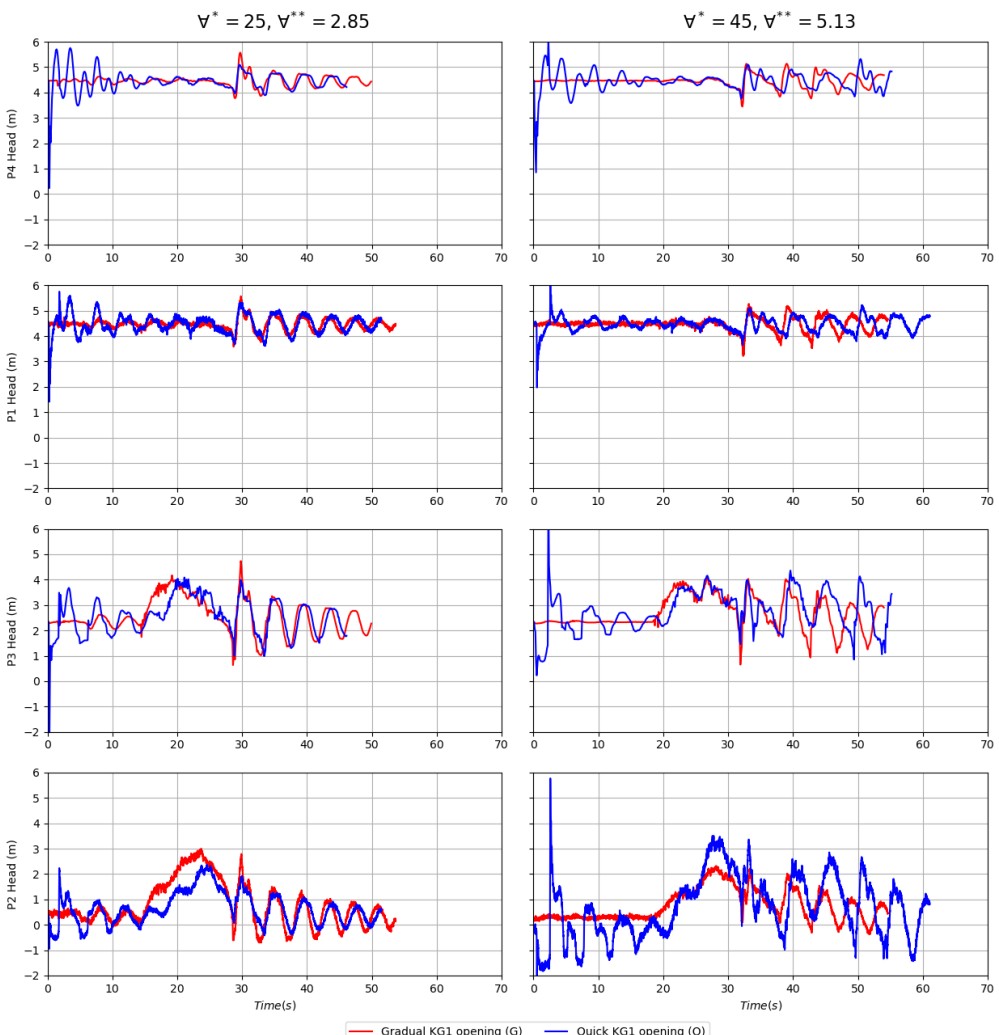

**Figure 7.** Pressure head results for each transducer for the case when $D^* = 0.50$, and higher initial water level in the water reservoir. Results in the left column are for the smaller air pocket, and the right column is for the larger air pocket volume.

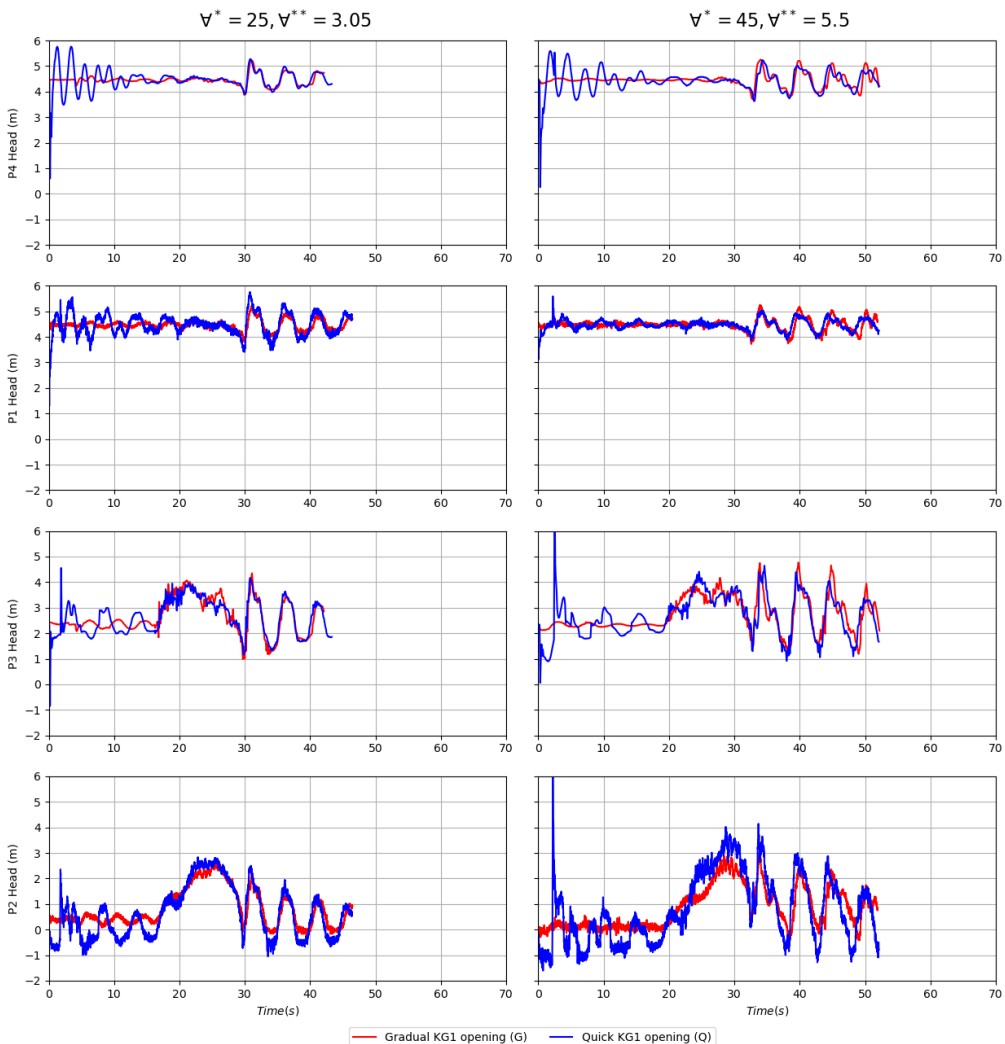

**Figure 8.** Pressure head results for each transducer for the case when $D^* = 0.50$, and lower initial water level in the water tank. Results in the left column are for the smaller air pocket, and the right column is for the larger air pocket volume.

## 4. Conclusions

This work presented the results of an experimental investigation of the release of entrapped air pockets through a water-filled vertical tower at the endpoint of a horizontal pipe. This condition is representative of upstream shafts in stormwater collection systems and tunnels, and such a geometric condition was not evaluated through experiments to date. Most experiments to date involved the release of air through an intermediate tower, which prevented the entire air phase mass to be released. The hypothesis of this research, which was confirmed through the measurements, is that the intensity of the geysering event increased.

The intensity of geysering was inferred in terms of the displacement of the water level within the tower. The free surface velocity in the tower was up to two to three times larger than the results presented by [11] for comparable tower diameters. The displacement of the free surface for $D^* = 1.0$ was also larger than the CFD results presented by [11]. Given that all cases in which $D^* = 0.50$ resulted in geysering, the total displacement during the air release was not measured for such cases. Regarding pressure measurements, pressure fluctuations were larger for the smaller-diameter tower and were impacted by the strategy of opening the knife gate valve that stored the air phase. Consistent with previous

studies, the piezometric head did not reach the observed water elevation during the air pocket release.

Finally, these experiments also demonstrated that the effect of a gradual release of the air pocket, which enabled time to avoid severe air–water surging during the tests, affected the quality of the measurements. The two-stage gradual opening enabled the pressurization of the air phase prior to its release in the apparatus, and this eliminated the pressure oscillations that are unrelated to geysering. We believe that future experiments evaluating strategies for air pocket release should apply this method to improve results. Other experimental conditions, changing the apparatus diameter and the geometry of the vertical shaft, should be performed in the future to corroborate the results presented here. Furthermore, additional experiments need to be performed to propose geysering control strategies in upstream shaft geometries similar to the case presented in this work.

**Author Contributions:** D.G.A., J.G.V., L.C.P. and R.T. contributed to the conceptualization of the study; D.G.A., J.G.V., L.C.P., R.T. and R.L.P. developed the methodology; J.G.V., L.C.P., C.G.P., B.M. and L.É.B. were responsible for conducting the investigation and validating the experimental data; L.C.P., L.É.B. and J.G.V. contributed to the visual presentation and formal data analyses; L.C.P., R.T., J.G.V. and R.L.P. red the original draft for submission; L.C.P., R.T., J.G.V., D.G.A., L.É.B. and R.L.P. revised and edited the paper; L.C.P., R.T., J.G.V. and D.G.A. supervised the work; R.T. and D.G.A. were responsible for the funding acquisition. All authors have read and agreed to the published version of the manuscript.

**Funding:** This work was supported partially by a National Science Foundation Grant No. 2048607. The authors would like to acknowledge the support of CAPES, which provided support for the second, sixth, and seventh authors, and to UFSM, which provided funding for experimental materials.

**Data Availability Statement:** All the data that support the findings of this study are available from the corresponding author upon reasonable request.

**Acknowledgments:** Present experimental runs used the facilities of the Laboratory of Environmental Engineering of the Federal University of Santa Maria (LEMA/UFSM) and the National Institute for Space Research (INPE/UFSM) in Santa Maria, Brazil.

**Conflicts of Interest:** The authors declare no conflicts of interest. The funders had no role in the design of the study, in the collection, the analyses, the interpretation of the data, in the writing of the manuscript, or in the decision to publish the results.

## Nomenclature

| Symbol | Unit | Description |
| --- | --- | --- |
| $D_t$ | m | Horizontal pipe internal diameter |
| $D$ | m | Vertical pipe internal diameter |
| $D^*$ | - | Normalized diameter |
| $g$ | $\mathrm{m \cdot s^{-2}}$ | Gravitational acceleration |
| $L_s$ | m | Vertical shaft length |
| $V_{FS}$ | $\mathrm{m \cdot s^{-1}}$ | Water free surface velocity |
| $V_{FS}^*$ | - | Normalized water free surface velocity |
| $Y_{FS,0}$ | m | Initial water free surface vertical coordinate |
| $Y_{FS,0}^*$ | - | Normalized initial water free surface vertical coordinate based on $L_s$ |
| $Y_{FS,0}^{**}$ | - | Normalized initial water free surface vertical coordinate based on $D$ |
| $\Delta Y_{FS}^*$ | - | Normalized water free surface displacement |
| $\Delta Y_{FS,max}^*$ | - | Maximum normalized water free surface displacement |
| $\forall$ | $\mathrm{m^3}$ | Air pocket volume |
| $\forall^*$ | - | Normalized air pocket volume based on $D_t$ |
| $\forall^{**}$ | - | Normalized air pocket volume based on the initial volume of water in dropshaft |

| Abbreviation | Meaning |
| --- | --- |
| CFD | Computational fluid dynamics |
| G | Gradual valve opening |
| HL | High water level |
| KG1, KG2 | Knife gate valve |
| LL | Low water level |
| PVC | Polyvinyl chloride |
| P1, P2, P3, P4 | Pressure transducer |
| Q | Quick valve opening |
| R1, R2, R3 | Repetition during experimental runs |
| T | Tee junction |

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
