# Peer review of "Experimental Study of Geysering in an Upstream Vertical Shaft"

_water, doi:10.3390/w15091740_

Round 1
Reviewer 1 Report
The authors presented new and valuable laboratory-testing data on the consequence of releasing a pocket of air into a pressurized/surcharged storm sewer tunnel. This laboratory study differs from previous studies in two major respects: (1) The simulated vertical shaft was located at the upstream end of a storm sewer network, so the entire pocket of the air had to move to and out of the vertical shaft, and (2) The air pocket was released both gradually and rapidly. The initial pressure inside the air pocket also differed from previous studies. However, the presentation can be improved before it is published. The examples of possible corrections or improvements follow:
(1) Title: The manuscript title indicates that the tested vertical shaft was located at an existing network's "downstream" end. But, throughout the main body of the manuscript, the tested vertical shaft was said to be located at the "upstream" end of the storm sewer network. The title should be consistent with the main body of the manuscript.
(2) Abstract, Lines 2-3: This sentence should be re-structured to more clearly differentiate between the cause of air pocket entrapment (extreme rain events and rapid inflows) and the consequence of the air pocket entrapment (abrupt pressure variations inside the systems).
(3) Abstract, Lines 12-14: Some words are missing at the end of the sentence. The complete sentence should have said that the measured pressure heads were much lower than the grade elevation.
(4) Line 183: Quantify the variability among the 3 to 6 tests under the same experimental conditions and indicate which test results were used in the later "Results and Discussion" section.
(5) Table 3. Separate the tests in the previous study (Muller et al. 2017) from those conducted in the present study. Maybe better to use different sequence numbers.
(6) Line 205: The English word "growth" may need to be replaced with another word, such as "rise." The word "growth" was also used later; the same replacement may be needed.
(7) Lines 303 to 305: This sentence needs to improve for clarity. The words "mitigate" and "not influence" seem conflicting in their meanings.
(8) References, Line 418-420: Delete this reference repeated (and numbered) later.
Author Response
The authors presented new and valuable laboratory-testing data on the consequence of releasing a pocket of air into a pressurized/surcharged storm sewer tunnel. This laboratory study differs from previous studies in two major respects: (1) The simulated vertical shaft was located at the upstream end of a storm sewer network, so the entire pocket of the air had to move to and out of the vertical shaft, and (2) The air pocket was released both gradually and rapidly. The initial pressure inside the air pocket also differed from previous studies. However, the presentation can be improved before it is published. The examples of possible corrections or improvements follow:
(1) Title: The manuscript title indicates that the tested vertical shaft was located at an existing network's "downstream" end. But, throughout the main body of the manuscript, the tested vertical shaft was said to be located at the "upstream" end of the storm sewer network. The title should be consistent with the main body of the manuscript.
- The authors would like to thank the reviewer for his/her consideration. We agree with the reviewer, and the title was corrected in the reviewed manuscript.
(2) Abstract, Lines 2-3: This sentence should be re-structured to more clearly differentiate between the cause of air pocket entrapment (extreme rain events and rapid inflows) and the consequence of the air pocket entrapment (abrupt pressure variations inside the systems).
- We agree with the reviewer that the sentence was unclear, and we reformulated it in the manuscript.
(3) Abstract, Lines 12-14: Some words are missing at the end of the sentence. The complete sentence should have said that the measured pressure heads were much lower than the grade elevation.
- The abstract was revised, and the issue is now fixed.
(4) Line 183: Quantify the variability among the 3 to 6 tests under the same experimental conditions and indicate which test results were used in the later "Results and Discussion" section.
- Maybe it was unclear in the initial manuscript, but the experimental runs were accomplished at least three times (repetitions with the same experimental configuration) to ensure reliability in the results. This number of runs was enough for most of the combinations assessed. Additional repetitions were done in cases of doubts related to the free surface displacement or operational problems (e.g., incorrect timing for the knife gate opening, pressure transducers not turned on). A small variability was observed among the repetitions, as shown in Figure 2. For instance, the graphics on the left side show filled circles which represent the three repetitions for the condition with a higher piezometric level, while the hollow circles display the three repetitions for the lower piezometric level condition. This sequence presents the results for the gradual (G) and quick (Q) knife gate opening, in addition to the air pocket volumes studied (25L and 45L). Aiming to clarify this point, we added the following sentence before presenting Table 3:
“Overall, minor differences in the results were verified for the same experimental conditions during the runs performed. Normally, the experimental repetitions were stopped in the third run (R1, R2, and R3); additional runs were performed either when inconsistent results were observed or due to operational problems during a given run.”
(5) Table 3. Separate the tests in the previous study (Muller et al. 2017) from those conducted in the present study. Maybe better to use different sequence numbers.
- Thanks for the suggestion. We coded the results by using a combination of letters and numbers. Thus, our study results were named as C1, C2, C3, etc., and Muller et al. (2017) results were coded as M1, M2, M3, and so on.
(6) Line 205: The English word "growth" may need to be replaced with another word, such as "rise." The word "growth" was also used later; the same replacement may be needed.
- We agree. Following this suggestion, the manuscript was revised.
(7) Lines 303 to 305: This sentence needs to improve for clarity. The words "mitigate" and "not influence" seem conflicting in their meanings.
- Thanks for the comment. We have reworded this sentence.
(8) References, Line 418-420: Delete this reference repeated (and numbered) later.
- It was corrected in the reviewed manuscript.
Reviewer 2 Report
General Comments
The paper presents an experimental setup and experiments showing the geysering effect in open pipe networks, similar to what is found in wastewater or draining networks. It is unclear what knowledge is gained from these experiments. It is clear that new data is created, but what knowledge does this bring? Does it show similarities with simulation results, or supporting a way of designing air relief systems? The authors should be clearer on this. For example, by stating hypothesis in the start of the paper, and conclude on these via the experiments. Maybe authors want to confirm the calculation in [8]?
The paper could be improved by including a symbol list.
In the introduction on page 2 third paragraph (marked 59-69), the contributions of the paper are described. Here it is stated that a gab in designing air relief systems exist. However, the paper does not specifically conclude on this. Likewise, numbers on the air catchment is given, but again there is no clear conclusion later in the paper.
The paper deals with a test setup for testing air relief capabilities. The tests should support the understanding of air relief systems in sewer networks, but the structure of the test setup is not compared to structure in real life sewers. To support the practical usability of the results, a discussion on the structure of the test setup compared to real life sewers should be included.
In the last paragraph of the introduction (marked 79-83), it would help the reader if section numbers for the different parts of the paper were mentions.
Along with Fig.1 that presents the test setup, it would be beneficial if comparisons to real sewer networks were added, answering questions such as: Where in the sewers are structure like this found? What weather phenomenon create the heysering effects and where in the world are they typically seen?
In the last part of 4 (marked 157-158), it is stated that air water surging where mostly absent. Maybe add a line on why this is important. Maybe compare to what is expected in real life sewers.
In Section 2.4 the experiment variables are explained. Please refer to Fig. 1 when explaining these. For example, in item a) a water tower is mentioned, but that are two tanks in Fig. 1? Likewise, the not normalized variable could be added to the figure, to help the reader to follow the experimental outcomes.
In the start of Section 3, (marked 203-220), please refer to the specific figures where the described phenomenon can be seen.
Please describe what R1-HL, R2_HL, etc. means in Figure 2.
When explaining sensor placements in Section 3.2 please refer to Figure 1.
It could be interesting to include a section in the conclusion with the authors opinion on how the results can be used from an engineering point of view.
Specific Comments
Abstract: In the second last line a sentence ends with and “the”. It seems that something is missing.
Figure 5, text: a higher initial water level is mention but higher than what?
Author Response
General Comments
The paper presents an experimental setup and experiments showing the geysering effect in open pipe networks, similar to what is found in wastewater or draining networks. It is unclear what knowledge is gained from these experiments. It is clear that new data is created, but what knowledge does this bring?
- The authors would like to thank the reviewer for his/her consideration. We have made changes in different locations in the manuscript to clearly state the research contributions, which we articulate in our response here. In essence, most past studies on geysering events triggered by uncontrolled air release used an intermediate vertical shaft to represent these events. Muller et al. (2017) showed that large fractions of the air pockets might not be released, indicating that geysering can be worse for a geometry like the one we used in this study. The last three paragraphs of the “Introduction and Objectives” section also discuss these knowledge gaps.
Does it show similarities with simulation results, or supporting a way of designing air relief systems? The authors should be clearer on this. For example, by stating hypothesis in the start of the paper, and conclude on these via the experiments. Maybe authors want to confirm the calculation in [8]?
- We have not performed numerical simulations of these geometries yet, so no comparison is made with simulations. No past studies have also considered using the CFD technique in the conditions we are using in this manuscript. Also, as is further explained ahead, the terminal/upstream dropshafts used in these tests are not representations of actual structures but rather are used to study this phenomenon experimentally. Once CFD results are available, which we hope will be soon, we can provide a better response regarding design recommendations for ventilation. We hope that this clarifies the reviewer’s question.
The paper could be improved by including a symbol list.
- Thanks for the suggestion. The symbol and abbreviation lists are now added to the manuscript.
In the introduction on page 2 third paragraph (marked 59-69), the contributions of the paper are described. Here it is stated that a gab in designing air relief systems exist. However, the paper does not specifically conclude on this. Likewise, numbers on the air catchment is given, but again there is no clear conclusion later in the paper.
- The reviewer may have misunderstood what we meant to achieve in this study. Most stormwater systems are not designed with ventilation as a parameter, but, de facto, any structure linking sewers and tunnels to atmospheric air provides ventilation. There is a knowledge gap not in the design of ventilation (since these are typically not considered in the design of stormwater systems), but rather a gap on how these vertical structures, once filled with water, behave as entrapped air pocket is released. This explains why the manuscript is not focusing on the design of ventilation structures but rather explores the flow characteristics as air is released.
The paper deals with a test setup for testing air relief capabilities. The tests should support the understanding of air relief systems in sewer networks, but the structure of the test setup is not compared to structure in real life sewers. To support the practical usability of the results, a discussion on the structure of the test setup compared to real life sewers should be included.
- Thanks for the suggestion. We have clarified in the manuscript that the apparatus has features resembling a stormwater system undergoing pressurization, but is not strictly equal to any existing stormwater system. As explained above, our goal is not to evaluate or test the ventilation of stormwater systems but rather to study the effects of uncontrolled air release in the geometry proposed in this study.
In the last paragraph of the introduction (marked 79-83), it would help the reader if section numbers for the different parts of the papers were mentions.
- Thanks for the suggestion. This has been implemented.
Along with Fig.1 that presents the test setup, it would be beneficial if comparisons to real sewer networks were added, answering questions such as: Where in the sewers are structure like this found? What weather phenomenon create the geysering effects and where in the world are they typically seen?
- As explained above, our apparatus was not meant to represent a real system geometry. This has been clarified in the present manuscript version.
In the last part of 4 (marked 157-158), it is stated that air water surging where mostly absent. Maybe add a line on why this is important. Maybe compare to what is expected in real life sewers.
- Thanks for the comment. The manuscript was amended to include, “Following the entrapment of air, the process of air pressurization should be gradual following the pressurization of water flows. Thus, the absence of surging prior to the uncontrolled release of air pockets is more representative.”
In Section 2.4 the experiment variables are explained. Please refer to Fig. 1 when explaining these. For example, in item a) a water tower is mentioned, but that are two tanks in Fig. 1? Likewise, the not normalized variable could be added to the figure, to help the reader to follow the experimental outcomes.
- Thanks for the suggestion. As we are introducing the variables, we now make a reference to Figure 1. Also, we changed the terminology for the vertical shaft so that this is not confounded with the water tank. We added the variables that were changed in the experiments in Figure 1 but not in the normalized fashion.
In the start of Section 3, (marked 203-220), please refer to the specific figures where the described phenomenon can be seen.
- The manuscript was amended to provide a more specific indication of the observed phenomena, as suggested.
Please describe what R1-HL, R2_HL, etc. means in Figure 2.
- R1, R2, and R3 represent the repetitions, while HL and LL represent the reservoir's high and low initial piezometric heads. This is now explained in Figure 2.
When explaining sensor placements in Section 3.2 please refer to Figure 1.
- The manuscript was amended as suggested.
It could be interesting to include a section in the conclusion with the authors opinion on how the results can be used from an engineering point of view.
- Thanks for your suggestions. However, at this point in the research, we believe it is premature to provide more specific recommendations in terms of the engineering design of these structures. More specific conclusions can be drawn as we perform the corresponding CFD research and come up with predictions of the system behavior when an upstream dropshaft discharges air pockets.
Specific Comments
Abstract: In the second last line a sentence ends with and “the”. It seems that something is missing.
- The abstract was restructured, and this issue was resolved.
Figure 5, text: a higher initial water level is mention but higher than what?
- It is now explained that this corresponds to the higher initial piezometric level in the reservoir, consistent with the terminology of the variables used in this study.
Reviewer 3 Report
The paper describes an interesting experimental campaign on sewer geysering and deserves publication after some minor reviews.
The abstract should synthesise the aim of the paper. Maybe you can sacrifice a phrase about the uniqueness of the experimental installation.
Figure 1 lettering should be larger.
Variables in Table 1 are all non-dimensional but some of them are ratios between incompatible units (mm^3/ L just to give you an example) that were declared before in the text. You should change the units or put a conversion value in the formulas.
Is it possible to have some practical conclusions aimed to address the design of attenuation measures in some critical configuration that can be found in engineering practice?
Author Response
The paper describes an interesting experimental campaign on sewer geysering and deserves publication after some minor reviews.
- Thanks for your comments.
The abstract should synthesize the aim of the paper. Maybe you can sacrifice a phrase about the uniqueness of the experimental installation.
- The abstract was restructured, and the goal of the manuscript is better explained.
Figure 1 lettering should be larger.
- Figure 1 has been edited according to your suggestions.
Variables in Table 1 are all non-dimensional but some of them are ratios between incompatible units (mm^3/ L just to give you an example) that were declared before in the text. You should change the units or put a conversion value in the formulas.
- Thanks for the remark. However, even though the quantities represented in the text are sometimes represented in units that are not SI (for instance, reservoir volume of 500 L) when these variables were represented in Table 3 (i.e., old Table 1), there was a conversion to SI. Thus there is no need for a conversion factor. For instance, the air pocket volume of 25 L (0.025 m3) was normalized by a diameter of 100 mm, or 0.1 m. This is represented in Table 3 as 0.025m3/(0.1m)^3=25.
Is it possible to have some practical conclusions aimed to address the design of attenuation measures in some critical configuration that can be found in engineering practice?
- Thanks for your comment, and we agree that a conclusion that could guide mitigation strategies would be good. However, this is the first group of experimental results using an apparatus of this kind, and at this point, we are simply attempting to understand what controls the severity of these geysering events. Our research on this problem is still ongoing using numerical tools, which would support more practical conclusions in terms of the attenuation approaches to this type of dropshaft geometry. Once CFD results are available, which we hope will be soon, we can provide a better response regarding design recommendations for ventilation. We hope that this clarifies the reviewer’s comment.
Round 2
Reviewer 1 Report
The authors understood my comments and suggestions very well and have fully addressed all of them in the revised version.
Reviewer 3 Report
The paper is well-structured and clear. The topic is fully in the aims of the journal and it is interesting for the audience. The experimental methodology is sound and generally the paper deserves publication in the present form